# Efficacy of a chemiluminescence-based urinary LAM (AIMLAM) assay in the diagnosis of active TB in Chinese children

Yiyi Chen,[1] Haiyan Li,[2] Jing Xiao,[1] Hui Qi,[1] Jie Kang,[1] Qifeng Li,[3] Jingjing Li,[1] Min Fang,[4,5] Li Duan,[4,5] Xiaomeng Wu,[1] Hailin Zhang,[2] Lin Sun[1]

**ABSTRACT** Child-friendly triage tests that do not rely on sputum for diagnosing active tuberculosis (ATB) in children are urgently required. This study aimed to assess a chemiluminescence-based urinary lipoarabinomannan test (AIMLAM) for diagnosing tuberculosis (TB) in children in China. Among the 579 children enrolled, 331 had TB, 169 had infectious diseases, and 79 were healthy controls. The area under the ROC curve of the AIMLAM test for distinguishing TB from non-TB (children with infectious diseases and healthy controls) was 0.74 (95% confidence interval [CI], 0.70–0.78), with a sensitivity of 52.2% (95% CI, 46.6%–57.6%) and a specificity of 91.9% (95% CI, 87.6%–94.9%). In 288 children with clinically diagnosed TB, the sensitivity of the test was 50.0% (95% CI, 44.1%–55.9%). LAM concentrations in extrapulmonary TB were significantly higher than those in pulmonary TB ($P = 0.003$), which resulted in a higher sensitivity of 63.6% (95% CI, 54.7%–72.1%). The diagnostic sensitivity of AIMLAM was significantly superior to that of Xpert ($P = 0.004$) and Ultra ($P = 0.002$). The sensitivity of AIMLAM is moderate but highest in extrapulmonary TB patients. The test appears to be promising for the rapid diagnosis of TB in children, especially in those with negative bacteriological results.

**IMPORTANCE** This study of a large-sample cohort of children in China assessed the diagnostic accuracy of the AIMLAM test kit as an auxiliary tool for childhood TB, and to determine an age-specific reference interval for LAM concentrations in children. Our findings showed that the overall sensitivity of AIMLAM was moderate, whereas a higher sensitivity was observed in patients with bacteriologically confirmed and severe TB. Age-specific cutoff values may be needed to optimize the diagnostic model in children. These results provide preliminary evidence for a method of diagnosing ATB in pediatric patients with samples that are difficult to obtain.

**KEYWORDS** pediatric tuberculosis, diagnostics, AIMLAM, joint diagnosis, clinical practice

Globally, approximately 10.8 million individuals experienced illness as a result of tuberculosis (TB) (1) in 2023. After interruptions to the diagnosis and treatment of TB caused by the coronavirus disease 2019 pandemic, the estimated number of fatalities attributable to TB in 2023 was reduced by 12% and 5.7% compared with 2021 and 2022, respectively. Nevertheless, the global community continues to fall well short of the World Health Organization's (WHO) End TB Strategy target of achieving a 50% reduction by 2025 (2).

In 2023, 12% of patients with TB were children and adolescents aged younger than 15 years. China ranks as the third largest country for the TB burden, with an estimated 90,000 children and adolescents affected (3). Children are at a considerable risk of rapidly advancing to severe TB because their immune systems are in the developmental stage. Consequently, an early and effective diagnosis for children is important to control TB in

**Peer Reviewers** Tabinda Anwar, Jinnah University for Women, Karachi, Pakistan; Andriansjah Andriansjah, Departemen of Microbiology, Medical Faculty, Universitas Indonesia, Jakarta, Indonesia

Address correspondence to Lin Sun, sunlinbch@163.com, or Hailin Zhang, zhlwz97@hotmail.com.

The authors declare no conflict of interest.

this at-risk population. Young children seldom have the capacity to produce sputum, and childhood TB is typically devoid of bacilli (2). Therefore, microbiological testing tends to show a relative lack of sensitivity.

Advances in the microbiological diagnosis of TB in children have made considerable progress over the past decade. The most important advancement is the development of molecular diagnostic tests, such as Xpert MTB/RIF (Xpert) (4) and Xpert MTB/RIF Ultra (Ultra) (5) (Cepheid, Sunnyvale, CA). These tests can rapidly identify *Mycobacterium tuberculosis* (MTB) and resistance to rifampicin. Although Xpert and Ultra have greatly improved the diagnostic accuracy of TB, instances of positives have been observed in repeated test results of previously treated patients because of residual nucleic acids (6). Additionally, the implementation of Xpert and Ultra in countries burdened by a high incidence of TB is constrained by factors such as cost, laboratory infrastructure (7), and challenges in obtaining suitable specimens for testing, particularly from young children. Therefore, the establishment of a child-friendly method for TB testing is imperative.

Lipoarabinomannan (LAM) is a crucial element of the MTB cell wall that can be metabolized by the kidneys and excreted in urine. The urine LAM test is a simple bedside procedure (8). LAM concentrations are correlated with bacterial load (9, 10) and are thus a reliable marker for active TB while mitigating the risk of false positives from residual nucleic acids in molecular diagnostics (11, 12). Measuring LAM can enhance the detection rate of ATB. This non-invasive urine sample collection method is especially beneficial for children, patients with extrapulmonary TB (e.g., urinary tract TB), human immunodeficiency virus (HIV)-infected individuals, older people, and those who struggle to provide sputum specimens (13, 14). The TB LAM test addresses the challenge of sample collection in traditional sputum tests by eliminating the need for coughing up samples. Furthermore, LAM remains stable in urine for 48 h at room temperature ($7^{\circ}$C–$22^{\circ}$C) (15), making it ideal for transport and testing in primary care settings.

Two main types of LAM kits are widely recognized. One of them is the Alere Determine TB-LAM Ag kit, which detects LAM antigen in urine by lateral flow immunochromatography. The WHO recommends the use of this kit in HIV-positive hospitalized patients (CD4 <200 cells/µL) (16), with a sensitivity of 42% (HIV-positive individuals with tuberculosis symptoms) and 35% (HIV-positive individuals without tuberculosis symptoms) (17). The other kit is the Fujifilm SILVAMP TB LAM kit, which detects silver-signal amplified conjugated high-affinity monoclonal antibodies. The detection accuracy of Fuji LAM was improved compared to Alere LAM. It has demonstrated an increased sensitivity of 70.4% in HIV-positive patients when assessed against the microbiological reference standard, and a sensitivity of 64.9% (18).

Recently, LDE Biosciences AIMLAM, a product designed for detecting LAM in urine, has enhanced sensitivity and efficiency while preserving high specificity through the use of high-affinity monoclonal antibody design, specialized sample pretreatment and concentration, and chemiluminescence techniques. Several studies have demonstrated the value of AIMLAM in the diagnosis of tuberculosis in adults. The chemiluminescence-based urinary LAM antigen demonstrated a sensitivity of 50.6% in adult tuberculosis (19) and a sensitivity of 83.3% in advanced HIV disease (13). High sensitivity was also observed in tuberculous pleural effusion (TPE) (49.5%) (20). Nevertheless, to the best of our knowledge, no studies have reported the utilization of AIMLAM in pediatric tuberculosis.

Therefore, this study of a large-sample cohort of children in China assessed the diagnostic accuracy of the AIMLAM test kit, which was included in the technology list that the WHO plans to review. We addressed the strategic need to "fill the evidence gap for childhood TB diagnosis" as outlined in the WHO Global TB Report 2024 (21). This study aimed to evaluate the diagnostic value of AIMLAM as an auxiliary tool for childhood TB and to determine an age-specific reference interval for LAM concentrations in children.

## MATERIALS AND METHODS

### Recruitment and classification of patients

From November 2021 to April 2025, children aged 18 years or younger were included in the evaluation of the AIMLAM (Guangzhou Leide Biosciences Co, Ltd) test if they had the following: (i) symptoms suggestive of TB (e.g., fever, cough, chest pain, night sweats, and weight loss) or imaging characteristics indicative of TB; (ii) completion of a clinical evaluation for TB (i.e., clinical radiographic examination) and laboratory testing (i.e., interferon-gamma release assay [IGRA], tuberculin skin test (TST), acid fast staining, MTB culture, Xpert MTB/RIF, or Xpert MTB/RIF Ultra; (iii) a definite diagnosis and complete clinical information; and (iv) no immunosuppression or immunodeficiency diseases. Patients who had previously undergone anti-TB treatment or who were being treated for more than 1 week were excluded from this study. Children diagnosed with infectious diseases who were admitted to the hospital concurrently and those who underwent a physical examination were enrolled as controls. The enrolled patients' demographic information and clinical data, such as age, sex, and a history of TB contacts, were collected from their medical records.

The enrolled children were finally classified as ATB, namely bacteriologically confirmed TB (BC-TB) (22) and clinically diagnosed TB (CD-TB) (23). BC-TB was defined as positive results of MTB culture or acid-fast staining of sputum or other samples. CD-TB was achieved on the basis of (i) clinical manifestations (e.g., fever, cough, chest pain, night sweats, and weight loss), (ii) positive laboratory results (a positive Xpert, Ultra, IGRA, or TST), (iii) imaging characteristics suggesting TB, and (iv) appropriate response to anti-TB therapy. Patients with ATB were then subclassified into the following two groups: (i) pulmonary TB cases, which were characterized by exclusive intrathoracic involvement (i.e., confined to the lung parenchyma, pleura, and intrathoracic lymph nodes) and (ii) extrapulmonary TB cases, which referred to participants with TB identified in organs or tissues outside the thorax, as well as those with concurrent pulmonary TB. Additionally, tuberculous meningitis, miliary TB, and disseminated TB were categorized as severe types of TB.

Children with infectious diseases were enrolled as the disease control (DC) group, which was defined as children who were symptomatic but did not fit the above-mentioned criteria and had confirmed evidence of viral, mycoplasma, or bacterial infections other than MTB. Additionally, follow-up showed improvement after treatment with antiviral drugs or antibiotics. In the healthy control (HC) group, we enrolled participants who exhibited no signs of chest radiological abnormalities, tested negative in IGRA and TST, were free of any infectious diseases, and showed no evidence of tuberculosis infection during follow-up. Children with incomplete information, HIV-positive results, and samples with high proteinuria and fatty urine were excluded from this study.

### Sample collection

Sputum, bronchoalveolar lavage, and urine were collected according to clinical practice standards within 3 days of hospitalization. Each specimen was transported to the laboratory within 6 h of collection. At least 10 mL of a clean midstream urine sample was collected for each participant. We selected the midstream urine and avoided using samples with high proteinuria and fatty urine (proteinuria was defined as urine protein levels exceeding 100 mg/L or 150 mg/24 h; fatty urine was cloudy urine with floating oil droplets, increased foam, free fat, oval fat bodies, and small fat tubular bodies).

### Culture and smear procedure

A total of 2 mL of respiratory specimen was thoroughly vortexed with 2 mL of 2% N-acetyl-L-cysteine sodium hydroxide and purified for 15–20 min. The resulting mixture was subsequently neutralized with sterile phosphate-buffered saline to achieve a final volume of 45 mL and centrifuged at 3,000 × $g$ for 15 min at 4℃. The sediment was resuspended in 1 mL of phosphate-buffered saline and transferred into the MGIT 960

system (Becton, Dickinson and Company, USA). The pellet was smeared onto a slide for Ziehl–Neelsen acid-fast staining and examined directly by microscopy.

## Xpert and Ultra assays

The Xpert and Xpert Ultra assays were carried out following the manufacturer's instructions. Briefly, 1 mL of the sputum was gently mixed with 2 mL of the sample reagent, vortexed for at least 10 s, then left to incubate at room temperature for 10 min. Afterward, the mixture was vortexed again for 10 s and incubated for another 5 min at room temperature. A total of 2 mL of this prepared mixture was transferred into the Xpert or Xpert Ultra cartridges, which were then loaded into the GeneXpert instrument.

## AIMLAM test procedures

LAM in urine was detected using a chemiluminescence immunoassay. A total of 100 µL of magnetic beads coated with LAM capture antibodies was added to 4 mL of urine in a sample tube. After the rotation reaction at room temperature, place it into a rotary mixer at a speed of 185 × $g$ for 2 h. A magnetic stand was used to adsorb the beads, and the supernatant was discarded. Following magnetic separation and washing, pre-excitation and excitation solutions were incorporated into the reaction mixture, which resulted in a LAM detection antibody immune complex comprising magnetic beads-antibody-antigen-acridine ester. The LAM content was directly proportional to the value of relative light units. The specific process is shown in the figure (Fig. S1). The instrument (SMART 500S, Chongqing Keysmile Biological Technology Co., Ltd.) automatically evaluated the relative light units produced by the sample against the values derived from the LAM calibration product to acquire the test results. The results were received after 30 min and classified as negative if <0.4 U/mL and positive if ≥0.4 U/mL. To reduce review bias, AIMLAM operators/readers were blinded to clinical classification.

## Statistical analysis

The statistical analysis was conducted using SPSS version 25.0 (IBM, Armonk, NY, USA) and Python 3.8.10 (Python Software Foundation, 2021). Sensitivity and specificity were compared using the chi-square test, while concordance among the various diagnostic tests was evaluated using the kappa value. Using McNemar's paired test to compare differences in sensitivity, specificity, PPV, and NPV between AIMLAM and Xpert/Ultra. The accuracy of AIMLAM was evaluated using an ROC curve, including the area under the ROC curve (AUC) and the best cutoff. The differences in AUCs between various groups were analyzed using the DeLong test. The best cutoffs were determined using the Youden index. Bootstrapping with optimism correction was employed to validate the new threshold, with a sampling size of 1,000. The optimism measure responded to the extent of overestimation (or underestimation) of the true performance of the uncorrected result. If optimism was <0.5%, it indicated that the uncorrected outcomes are stable and devoid of optimism bias. A $P$ value <0.05 was considered statistically significant.

## RESULTS

### Clinical characteristics of the study population

A total of 579 participants provided urine samples and the AIMLAM test was performed. These participants comprised 331 children with ATB (median age: 7.4 years, interquartile range [IQR]: 4.0–11.1), 169 with infectious diseases (DCs) (median age: 5.3 years, IQR: 2.9–8.2), and 79 HCs (median age: 4.9 years, IQR: 3.1–5.8) (Fig. 1). Among all of the participants (median age: 6.4 years, IQR: 3.3–9.8), 11.7% (68/579) reported a history of TB contact, 50.8% (294/579) had positive TST results, and 298 (51.5%) had positive IGRA results. Among the 331 children diagnosed with ATB, 43 were classified as BC-TB and 288 as CD-TB. In the DC group, 25(14.8%) patients had combined latent TB infection (LTBI) based on a positive IGRA and TST results (Table 1).

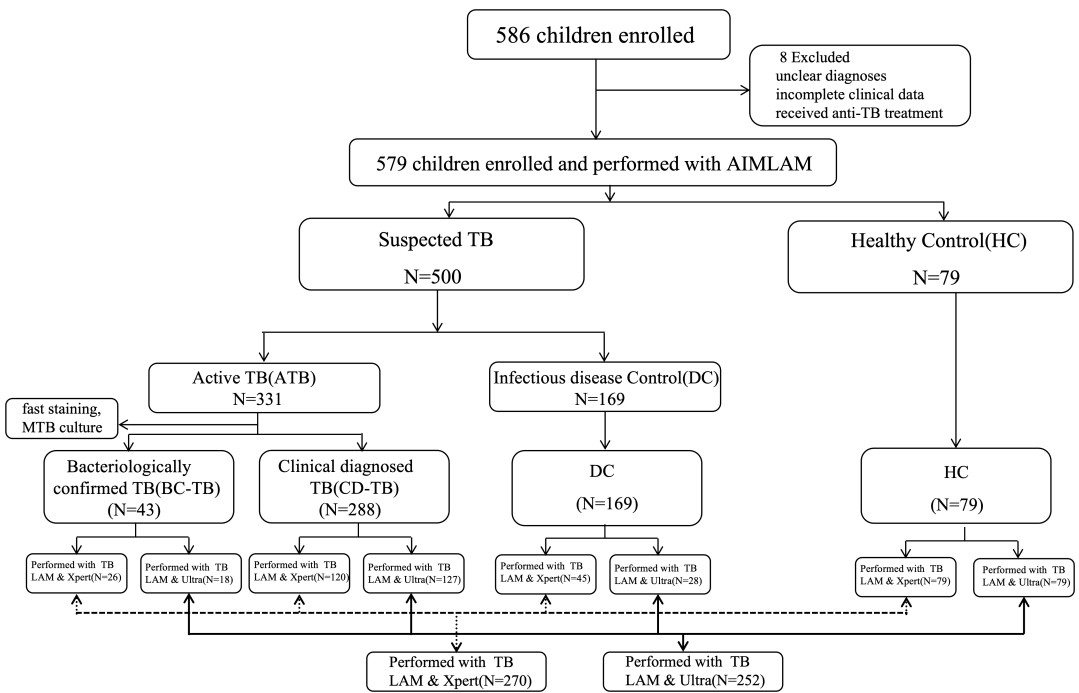

**FIG 1** Flow chart of the study population. TB, tuberculosis; DCs, infectious disease control; HC, healthy control; BC-TB, bacteriologically confirmed TB; CD-TB, clinically diagnosed TB; AIMLAM, chemiluminescence-based urinary LAM; Xpert, Xpert MTB/RIF; and Ultra, Xpert MTB/RIF Ultra.

## Performance of AIMLAM in the diagnosis of TB in children

### Sensitivity and specificity of AIMLAM

A Sankey diagram (Fig. 2) showed that of 193 (33.3%, 193/579) positive AIMLAM results, 29 were attributed to BC-TB, 144 to CD-TB, and 20 to the DC (80.0%, 16/20) and HC (20.0%, 4/20) groups. The AIMLAM test showed a sensitivity of 52.2% (95% confidence interval [CI], 46.6%–57.6%) in all of the 331 patients with TB and a specificity of 91.9% (95% CI, 87.6%–94.9%) in the 248 control subjects (DC and HC groups). A higher sensitivity was observed in children with BC-TB than in those with CD-TB (67.4%; 95% CI, 51.3%–80.5% vs. 50.0%; 95% CI, 44.1%–55.9%) ($P = 0.049$). Significant variations in sensitivity were observed across different age groups, with a sensitivity of 62.9% (95% CI, 50.4%–73.9%) in children younger than 5 years of age, 46.8% (95% CI, 37.4%–56.5%) in those 5–10 years of age, and 51.3% (95% CI, 43.1%–59.5%) in those older than 10 years of age. The lowest specificity was found in the youngest age group, with a specificity of 88.0% (95% CI, 79.2%–93.6%). The sensitivity of AIMLAM was significantly higher in severe TB cases (69.4%; 95% CI, 54.4%–81.3%) than in mild TB cases (49.3%; 95% CI, 43.5%–55.5%) ($P = 0.011$). The AIMLAM test showed a significantly greater diagnostic sensitivity with extrapulmonary TB (63.6%; 95% CI, 54.7%–72.1%) than with pulmonary TB (46.0%; 95% CI, 39.2%–52.9%) ($P = 0.003$) (Table 2).

### Best cutoff value of AIMLAM in the diagnosis of TB in children

The optimal thresholds of AIMLAM were then analyzed to achieve the best diagnostic accuracy in children. The optimal positive thresholds for discriminating total ATB from non-TB controls were 0.1, 0.13, and 0.14, respectively (Fig. 3A through G). First, the DC and HC groups were merged as the non-TB controls. A moderate accuracy of AIMLAM was observed in diagnosing the total ATB, with an AUC of 0.74 (95% CI, 0.70–0.78). A similar diagnostic accuracy was observed in BC-TB (AUC, 0.82; 95% CI, 0.72–0.91) and CD-TB (AUC, 0.74; 95% CI, 0.69–0.79), respectively (Fig. 3A through C). Either comparing

**TABLE 1** Main clinical characteristics of the study population

| Characteristics | Total $N = 579$, n (%) | Active tuberculosis (ATB) $N = 331$, n (%) | Non-tuberculosis (non-TB) ($N = 248$) | |
|---|---|---|---|---|
| | | | Infectious disease control (DC) $N = 169$, n (%) | Healthy control (HC) $N = 79$, n (%) |
| Age (in years) | | | | |
| Mean (interquartile range) | 6.4 (3.3–9.8) | 7.4 (4.0–11.1) | 5.3 (2.9–8.2) | 4.9 (3.1–5.8) |
| Full range | 0.3–16.3 | 0.3–15 | 0.3–14 | 2–14 |
| Gender | | | | |
| Male | 348 (60.1) | 192 (57.9) | 103 (61.2) | 53 (67.1) |
| Female | 231 (39.9) | 139 (42.1) | 66 (38.8) | 26 (32.9) |
| History of tuberculosis contacts | | | | |
| Yes | 68 (11.7) | 53 (16.0) | 16 (9.5) | 0 (0) |
| No | 432 (74.6) | 226 (68.3) | 136 (80.5) | 70 (88.6) |
| Unknown | 79 (13.6) | 52 (15.7) | 18 (10.0) | 9 (11.4) |
| Tuberculin skin test | | | | |
| Positive | 294 (50.8) | 279 (84.3) | 15 (8.9) | 0 (0) |
| Negative | 180 (31.1) | 43 (13.0) | 137 (81.1) | 79 (100.0) |
| No data | 105 (18.1) | 9 (2.7) | 17 (10.0) | 0 (0) |
| Interferon-γ release assay | | | | |
| Positive | 298 (51.5) | 288 (87.0) | 10 (5.9) | 0 (0) |
| Negative | 153 (26.4) | 38 (11.5) | 115 (68.0) | 79 (100.0) |
| No data | 128 (22.1) | 5 (1.5) | 44 (26.1) | 0 (0) |

with healthy children (Fig. 3D through F) or those with infectious diseases (Fig. 3G through I), the diagnostic accuracy of AIMLAM was similar in diagnosing ATB.

The diagnostic accuracy metrics of AIMLAM distinguishing ATB from Non-TB were further validated at thresholds of 0.14, 0.13, and 0.1 using Bootstrapping. The corrected sensitivities were 67.4% (95% CI, 62.2%–72.4%), 68.5% (95% CI, 63.3%–73.7%), and 70.8% (95% CI, 65.4%–76.0%), respectively. The specificity values were 80.6% (95% CI, 75.5%–85.2%), 79.5% (95% CI, 74.4%–84.7%), and 77.5% (95% CI, 71.9%–82.5%), respectively (Table S1 and S5).

## Effectiveness of AIMLAM across various TB subgroups

Among the 331 children with ATB, 49 had severe TB and 282 had mild TB. A violin plot showed that data density in the mild TB group was mainly observed in the low value regions and showed a tighter distribution than that in the severe group. The median concentration of AIMLAM in the mild group was 0.28 U/mL (IQR: 0.10–0.86), which was lower than that (0.68 U/mL, IQR: 0.16–2.21) in the severe group. Nevertheless, this difference was not significant, which may have been due to more dispersed data in the severe group (Fig. 4A). In addition, 213 children had pulmonary TB, and 118 were diagnosed with extrapulmonary TB. The median AIMLAM concentration in children with extrapulmonary TB (0.85, IQR: 0.16–2.17 U/mL) was significantly higher than that in those with pulmonary TB (0.47, IQR 0.1–2.17 U/mL) ($P = 0.013$, Fig. 4B).

Among the 118 children with extrapulmonary TB, the main types were abdominal TB and tuberculous peritonitis (44.1%) and tuberculous meningitis (33.9%) (Fig. 4C). Abdominal TB and tuberculous peritonitis frequently presented in conjunction with other forms of extrapulmonary TB, which contributed to the elevated detection rate with AIMLAM (37/52, 71.2%) (Fig. 4D). The sensitivity of AIMLAM for tuberculous meningitis and bone TB was similar (65%, 26/40 vs. 73.3%, 11/15). All five cases with urinary tract TB had positive results (100.0%). By contrast, tuberculous pericarditis (40.0%, 4/10) and lymph node TB (46.7%, 7/15) showed the lowest detection rates.

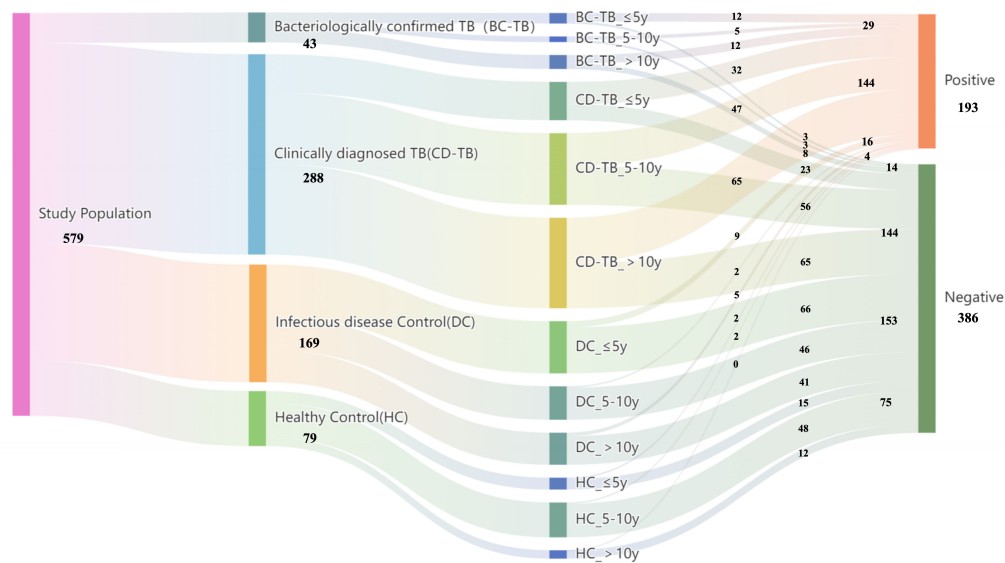

**FIG 2** Sankey diagram of the distribution of population flows for AIMLAM diagnosis.

To compare the differences between AIMLAM and molecular tests in diagnosing extrapulmonary and pulmonary TB, we also examined the diagnostic performance of Xpert and Ultra in intrapulmonary and extrapulmonary cases (Table S2). The data showed that, compared to pulmonary TB (27.8%; 95% CI, 18.6%–39.2%), Ultra demonstrated a much higher sensitivity for extrapulmonary TB at 47.0% (95% CI, 34.7%–69.6%). Likewise, Xpert's sensitivity for extrapulmonary TB was 38.3% (95% CI, 24.9%–53.6%), which was more than twice as high as its sensitivity for pulmonary TB at 13.1% (95% CI, 7.5%–21.8%).

## Concordance between AIMLAM and molecular tests

A total of 270 children (146 patients with ATB and 124 non-TB patients) were tested simultaneously with AIMLAM and Xpert. The overall concordance was 0.400 (Fig. 5A, Table 3). The sensitivity of AIMLAM and Xpert was 46.6% (95% CI, 38.7%–54.7%) and 21.2% (95% CI, 15.4%–28.6%), respectively (Table 3). In patients with BC-TB, the sensitivities of AIMLAM and Xpert were similar (both were 50.0%), with a kappa value of 0.505 (Fig. 5B). However, in patients with CD-TB, AIMLAM detected significantly more positive cases than that with Xpert (45.8%, 55/120 vs. 15.0%, 18/120) ($P = 0.001$), which resulted in a moderate kappa value of 0.456. Six cases with Xpert-negative but IGRA-positive results were found to be AIMLAM-positive (Fig. 5C, Table 3). AIMLAM successfully identified 37 Xpert-negative cases of CD-TB, which resulted in a higher sensitivity of 45.8% (95% CI, 37.2%–54.7%) (Fig. 5C, Table 3). Considering the limitation of Xpert performance in smear-positive samples (24), we conducted a comparison of the LAM performance and Xpert performance in both smear-positive and smear-negative samples. The results indicated that LAM exhibited higher sensitivity in both categories, with rates of 68.8% and 52.6%, respectively (Table S3).

A total of 252 children (145 patients with ATB and 107 non-TB patients) were tested simultaneously with AIMLAM and Ultra. The concordance of the two tests for the diagnosis of ATB was 0.518 (Fig. 5D, Table 3). The sensitivity of AIMLAM was significantly greater than that of Ultra (55.9%, 95% CI, 47.7%–63.7% vs. 37.2%; 95% CI, 29.8%–45.3%) ($P = 0.002$). In patients with BC-TB, the sensitivities of AIMLAM and Ultra were similar (77.8% vs. 88.3%), with a high kappa value of 0.826 (Fig. 5E). However, in patients with CD-TB, AIMLAM detected significantly more positive cases than Ultra (52.8%, 67/127 vs. 30.7%, 39/127) ($P < 0.001$), which resulted in a moderate kappa value of 0.498. Ultra only

TABLE 2 The accuracy of AIMLAM in children categorized by age, diagnostic criteria, disease severity, and location of occurrence[a]

| Group | Sensitivity% (95% CI), n/N | P value | PPV% (95% CI), n/N | P value | NPV% (95% CI), n/N | P value | Specificity% (95% CI), n/N | P value |
|---|---|---|---|---|---|---|---|---|
| Active TB (N = 331) | 52.2 (46.6–57.6),173/331 | | 89.6 (83.6–93.0), 173/193 | | 59.1 (54.0–64.0), 228/386 | | 91.9 (87.6–94.9), 228/248 | |
| ≤5 years | 62.9 (50.4–73.9), 44/70 | <0.001 | 80.0 (66.7–89.1), 44/55 | 0.938 | 75.7 (66.2–83.2), 81/107 | 0.868 | 88.0 (79.2–93.6), 81/92 | 0.940 |
| 5–10 years | 46.8 (37.4–56.5), 52/111 | | 92.9 (81.9–97.7), 52/56 | | 61.4 (53.2–69.1), 94/153 | | 95.9 (89.3–98.7), 94/98 | |
| >10 years | 51.3 (43.1–59.5), 77/150 | | 93.9 (85.7–97.7), 77/82 | | 42.1 (33.4–51.2), 53/126 | | 91.4 (80.3–96.8), 53/58 | |
| Bacteriologically confirmed TB (BC-TB) (N = 43) | 67.4 (51.3–80.5), 29/43 | 0.049 | 59.2 (43.3–71.5), 29/49 | <0.001 | 94.2 (90.3–96.7), 228/242 | 0.557 | | |
| Clinically diagnosed TB (CD-TB) (N = 288) | 50.0 (44.1–55.9),144/288 | | 87.8 (89.0–91.8),144/164 | | 61.3 (56.1–66.2), 228/372 | | | |
| Severe TB (N = 49) | 69.4 (54.4–81.3),34/49 | 0.011 | 62.9 (48.7–75.4), 34/54 | <0.001 | 93.8 (89.8–96.4), 228/243 | <0.001 | 91.9 (87.6–94.9), 228/248 | 1.000 |
| Mild TB (N = 282) | 49.3 (43.3–55.5),139/282 | | 87.4 (81.0–92.0),139/159 | | 61.5 (56.3–66.4), 228/371 | | | |
| Pulmonary TB (N = 213) | 46.0 (39.2–52.9), 98/213 | 0.003 | 83.1 (74.8–89.1), 98/118 | 0.558 | 66.5 (61.2–71.4), 228/343 | <0.001 | | |
| Extrapulmonary TB (N = 118) | 63.6 (54.1–72.1), 75/118 | | 78.9 (69.1–86.3), 75/95 | | 84.1 (79.1–88.2), 228/271 | | | |

[a]CI, confidence interval; PPV, positive predictive value; NPV, negative predictive value.

identified one AIMLAM-negative case as positive, with a specificity of 99.1% (95% CI, 94.9%–99.8%) (Fig. 5F, Table 3).

## Advantages of combined methods for diagnosing TB in children

The combined use of AIMLAM and Xpert for diagnosing TB raised the diagnostic sensitivity to 56.2% (82/146) (Table S4), which was nearly three times higher than the sensitivity observed with Xpert alone (21.2%, 31/146). The combined use of AIMLAM and Ultra to diagnose TB showed an even greater sensitivity of 64.1% (93/145) (Table S4), which was nearly double that of using only Ultra for a diagnosis (37.2%, 54/145) (Table 3, Fig. 5F).

Furthermore, compared to culture (sensitivity of 9.4% [95% CI, 6.5%–13.2%] and specificity of 100% [95% CI, 98.1%–100.0%]), the combination of AIMLAM and any of the molecular tests significantly increased diagnostic sensitivity.

## DISCUSSION

Enhancing detection of TB in patients and broadening treatment options are vital for preventing transmission and the progression of severe disease. Therefore, a new child-friendly diagnostic method with high sensitivity is urgently required. To meet this requirement, we conducted a diagnostic study to evaluate the effectiveness of a urine-based test for detecting TB in children in a highly endemic country. We found that the AIMLAM test showed a notable diagnostic performance in children. Furthermore, our study indicated the considerable benefit of AIMLAM regarding diagnostic sensitivity among these participants and its critical function in identifying extrapulmonary TB.

Childhood TB is usually misdiagnosed as pneumonia, systemic bacterial and viral infections, or malnutrition because of its nonspecific clinical manifestations (25). In the present study, the AIMLAM test detected 50.0% of TB cases with negative bacteriological results. This offers a more accurate basis for clinical treatment. It helps clinicians

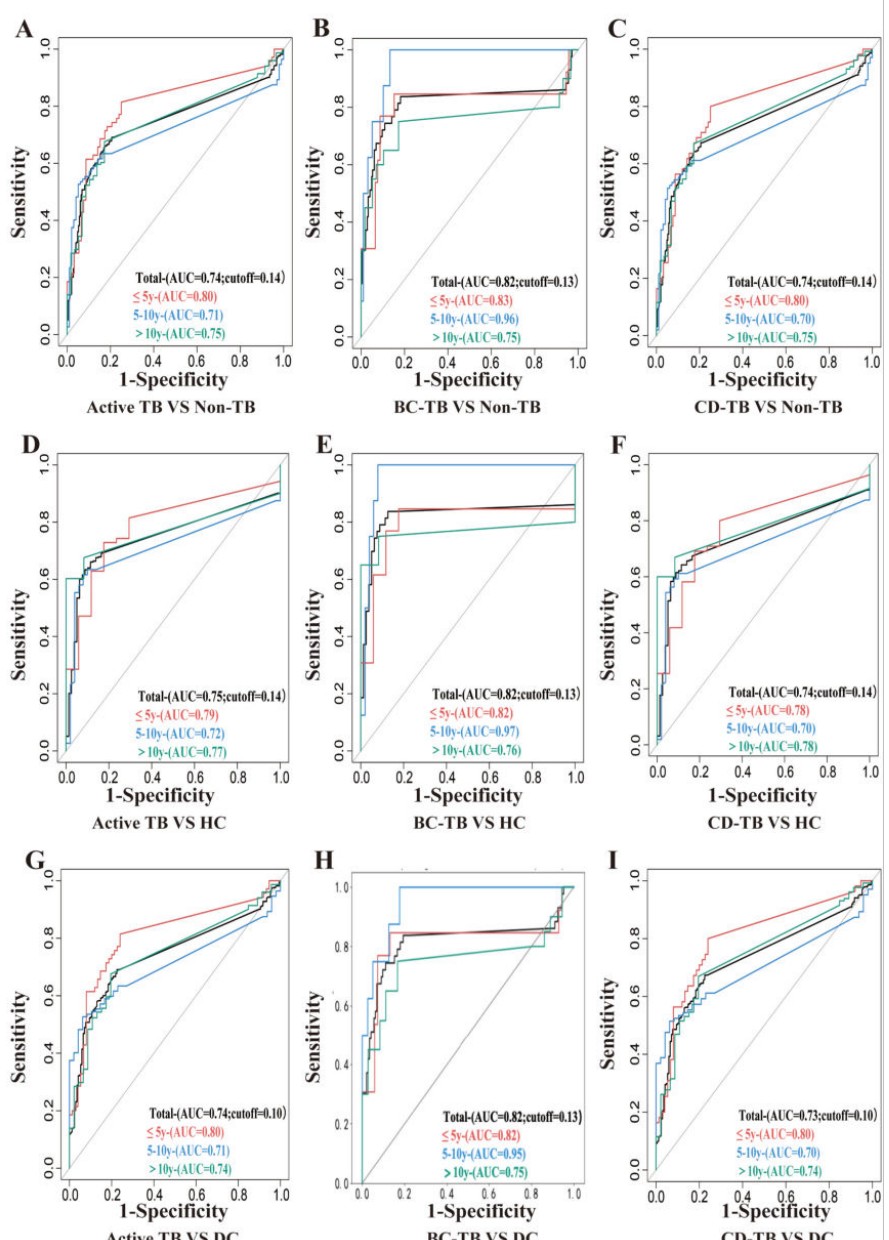

**FIG 3** ROC curve of AIMLAM for the detection of TB. (A–C) ROC curves of AIMLAM in different age groups to distinguish between non-TB and TB populations (A: Active TB, B: BC-TB, and C: CD-TB), including the AUCs and the overall optimal cutoff value; (D–F) ROC curves of AIMLAM in different age groups to distinguish between TB populations (A: Active TB, B: BC-TB, and C: CD-TB) and HC populations, including the AUCs and the overall optimal cutoff value; (G–I) ROC curves of AIMLAM in different age groups to distinguish between TB populations (A: Active TB, B: BC-TB, and C: CD-TB) and DC populations, including the AUCs and the overall optimal cutoff value.

recognize the possibility of tuberculosis earlier and prevents delays in treatment caused by misdiagnosis or the progression of severe illness due to missed diagnosis. Additionally, AIMLAM demonstrated superior sensitivity within the population with either severe TB (69.4%) or extrapulmonary TB (63.6%). Identifying severe and extrapulmonary TB presents great challenges because of the diverse clinical manifestations, which are contingent upon the affected organ, the stage of the disease, and the immune response of the host (20). Direct evidence of extrapulmonary TB is defined by a positive microbiological result obtained from an invasive specimen of suspected lesions, which

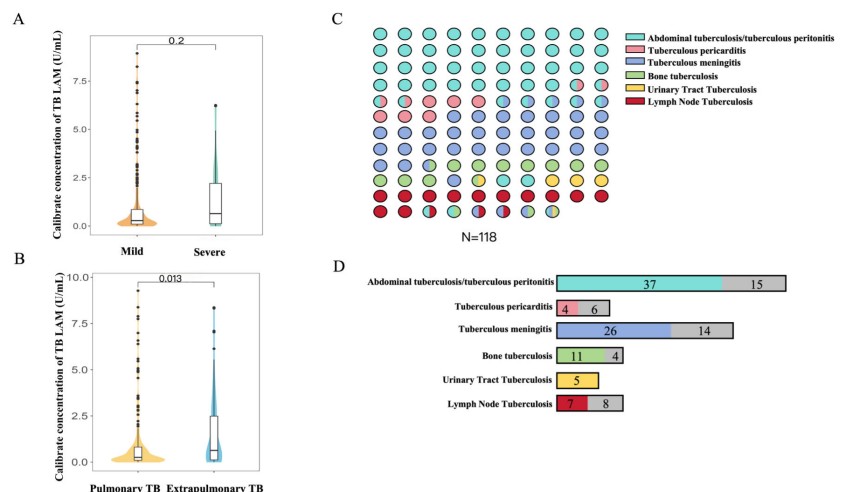

**FIG 4** Comparison of AIMLAM test results in different subgroups. (A) Violin plot of the difference in the diagnostic presentation of AIMLAM in mild TB and severe TB. (B) Violin plot of the difference in the diagnostic presentation of AIMLAM in pulmonary TB and extrapulmonary TB. (C) Distribution of different types of extrapulmonary tuberculosis. (D) Diagnostic efficacy of AIMLAM in different extrapulmonary tuberculosis. Colored squares represented LAM-positive results, and gray squares represented LAM-negative results.

are difficult to obtain and paucibacillary. Therefore, negative results are common in children with severe TB or extrapulmonary TB (26). The sensitivity of AIMLAM for celiac TB, bone TB, and tuberculous meningitis was high. LAM is a type of immunogenic lipopolysaccharide and is released from metabolically active or degenerating bacterial cells. Therefore, disseminated TB leads to higher antigen concentrations of LAM in the peripheral circulation and urine. Notably, in this study, AIMLAM showed a high sensitivity for urinary tract TB, with all five cases testing positive. Nevertheless, the number of cases considered was limited, necessitating a larger sample size for additional validation.

AIMLAM showed a significantly superior diagnostic performance when compared with Xpert and Ultra. In BC-TB cases, the diagnostic performance of AIMLAM is consistent with that of Xpert and Ultra, allowing it to function as a complement to diagnostic criteria. Combining AIMLAM with Xpert improved the sensitivity to 56.2%, while the AIMLAM and Ultra combination achieved a sensitivity of 64.1%, which substantially increased the diagnostic accuracy over Xpert or Ultra individually. Based on the low cost of the AIMLAM kit, approximately 14 dollars (100 RMB) for each test, it can be used as a screening test in children with suspected symptoms, especially in resource-constrained settings. From the perspective of efficient health economics, the combined approach proved to be more effective due to "complementary effects" (e.g., rapid AIMLAM screening followed by precise confirmation with Xpert or Ultra), which contributed to reducing hidden costs associated with ineffective testing and misdiagnoses while improving resource utilization. This combined approach can avoid the risk of high costs in a single diagnosis and reduce the resource pressure of the health system. In the future, this test can be further combined with the allocation of resources, such as testing costs and labor, to optimize the implementation pathway of combined strategies.

On the basis that heterogeneity exists among different populations, establishing varying positive thresholds for pediatric patients with TB is essential to more effectively identify the affected group. By balancing sensitivity and specificity, we concluded that a cutoff of 0.14 U/mL (sensitivity of 67.4% and specificity of 80.6%) is more appropriate for diagnosing tuberculosis in children. Overall, the threshold identified in the present study for diagnosing ATB (cutoff = 0.14) was lower than the positive threshold specified in the kit (cutoff = 0.4). Potential reasons for this include the typically lower bacterial load in children infected with tuberculosis compared to adults. Consequently, using adult

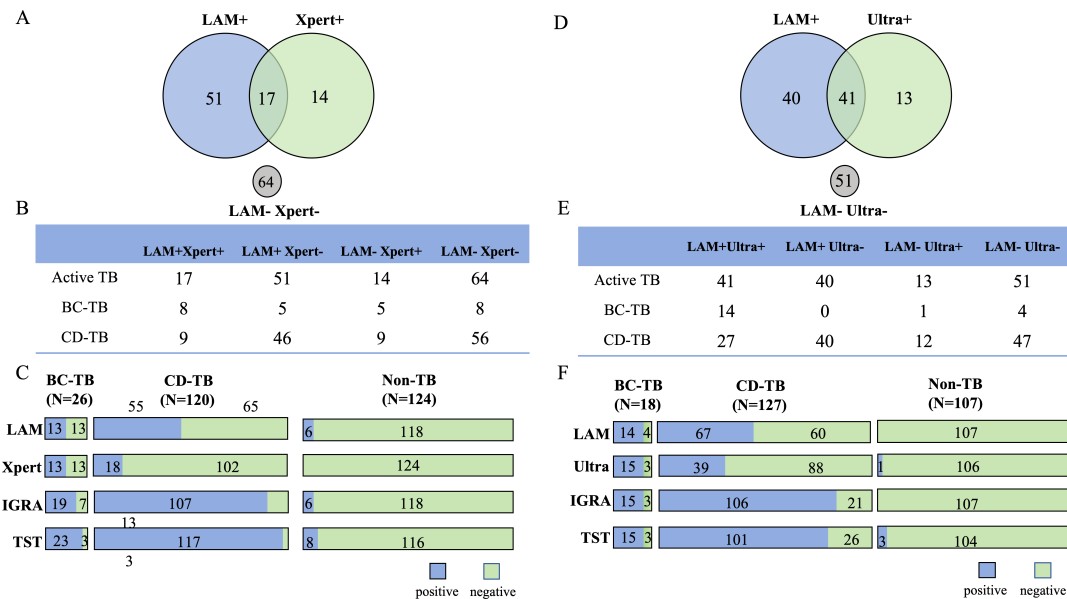

**FIG 5** Performance of different assays of AIMLAM and other assays. (A) Venn diagram of AIMLAM and Xpert in participants classified as active TB (N = 270). (B) Table of AIMLAM and Xpert in participants classified as active TB, BC-TB, and CD-TB (N = 270). (C) Results profile of the detection of 270 participants with different TB classifications via AIMLAM, Xpert, IGRA, and TST. (D) Venn diagram of AIMLAM and Ultra in participants classified as active TB (N = 252). (E) Table of AIMLAM and Xpert in participants classified as active TB, BC-TB, and CD-TB (N = 252). (F) Results profile of the detection of 252 participants with different TB classifications via AIMLAM, Ultra, IGRA, and TST.

high thresholds will lead to a high number of false-negative results in children. However, larger sample sizes or multicenter studies are needed to verify this finding.

In non-TB cases, 20 positive AIMLAM results were identified, with 16 from patients with lung infections and 4 from healthy individuals. Among the 16 cases with lung infections, 2 were positive for the IGRA and 2 for the TST, which may indicate latent infections. However, all 16 cases of infectious disease showed clear signs of bacterial and viral infection and were discharged without anti-tuberculosis therapy. Follow-up revealed no active tuberculosis in any of the 20 cases. Therefore, we consider these 20 cases likely false positives, possibly caused by cross-reactivity, such as with non-tuberculous mycobacteria (NTM).

Nonetheless, our study has certain limitations. We did not conduct a comparative analysis of the detection accuracy between AIMLAM and Alere Determine TB-LAM and/or Fujifilm SILVAMP TB-LAM under the same operating conditions. The absence of a horizontal comparison among similar detection techniques constitutes a limitation of our study. In addition, given the growing burden of pediatric multidrug-resistant tuberculosis (MDR-TB), we attempted to analyze the accuracy of AIMLAM detection in MDR-TB populations. In the present study, a total of six children were confirmed to be rifampicin-resistant by Xpert and Ultra. The sample size for rifampicin-resistance statuses was limited for subsequent analysis. A recent study revealed that Mannose-capped lipoarabinomannan (ManLAM)-related genes (pimB, mptA, dprE1, and embC) were most up-regulated under isoniazid (INH) treatment, indicating their involvement in drug resistance and MTB adaptability through ManLAM modulation (25). This suggests that the detection of AIMLAM or its related genes might play a role in the diagnosis of MDR-TB, which needs to be further validated with a large sample size. Furthermore, although NTM infection rates were low, we did not include an NTM disease group to avoid cross-reactivity influence. Finally, as a "Strict reference standard (SRS)" including "response to anti-TB therapy," which is susceptible to confirmation bias, we enrolled BC-TB populations that were eligible for positive bacteriologic tests and did not include populations for whom response to anti-TB therapy only. Therefore, further studies with

**TABLE 3** Comparison of diagnostic efficacy between AIMLAM and Xpert, and between AIMLAM and Ultra for pediatric tuberculosis

| Performance measurement | AIMLAM vs Xpert | | P value[a] | Kappa value | AIMLAM vs Ultra | | P value[b] | Kappa value |
|---|---|---|---|---|---|---|---|---|
| | AIMLAM | Xpert | | | AIMLAM | Ultra | | |
| Active TB | | | | | | | | |
| Sensitivity % | 46.6 | 21.2 | 0.004 | | 55.9 | 37.2 | 0.002 | |
| (95% CI), *n/N* | (38.7–54.7), 68/146 | (15.4–28.6), 31/146 | | | (47.7–63.7), 81/145 | (29.8–45.3), 54/145 | | |
| Specificity% | 95.2 | 100.0 | 0.029 | | 100.0 | 99.1 | 1.000 | |
| (95% CI), *n/N* | (89.8–97.8), 118/124 | (97.0–100.0), 124/124 | | 0.4 | (96.5–100.0), 107/107 | (94.9–99.8), 106/107 | | 0.518 |
| PPV% | 91.9 | 100.0 | 0.176 | | 100.0 | 98.2 | 0.404 | |
| (95% CI), *n/N* | (83.6–96.2), 68/74 | (89.0–100.0), 31/31 | | | (95.5–100.0), 81/81 | (90.4–99.7), 54/55 | | |
| NPV% | 60.2 | 51.9 | 0.099 | | 62.6 | 53.8 | 0.092 | |
| (95% CI), *n/N* | 53.2–66.8), 118/196 | (45.6–58.1), 124/239 | | | (55.1–69.5), 107/171 | (46.8–60.6), 106/197 | | |
| Bacteriologically confirmed TB | | | | | | | | |
| Sensitivity % | 50.0 | 50.0 | 1.000 | | 77.8 | 83.3 | 1.000 | |
| (95% CI), *n/N* | (32.1–67.9), 13/26 | (32.1–67.9), 13/26 | | | (54.8–91.0), 14/18 | (60.8–94.2), 15/18 | | |
| Specificity% | 95.2 | 100.0 | 0.05 | | 100.0 | 99.1 | 1.000 | |
| (95% CI), *n/N* | (89.8–97.8), 118/124 | (97.0–100.0), 124/124 | | 0.505 | (96.5–100), 107/107 | (94.9–99.8), 106/107 | | 0.826 |
| PPV% | 68.4 | 100.0 | 0.059 | | 100.0 | 93.8 | 1.000 | |
| (95% CI), *n/N* | (46.0–84.6), 13/19 | (77.2–100.0), 13/13 | | | (78.5–100.0), 14/14 | (71.7–98.9), 15/16 | | |
| NPV% | 90.1 | 90.5 | 1.000 | | 96.4 | 97.2 | 1.000 | |
| (95% CI), *n/N* | (83.8–94.1), 118/131 | (84.4–94.4), 124/124 | | | (91.1–98.6), 107/111 | (92.2–99.1), 106/109 | | |
| Clinically diagnosed TB | | | | | | | | |
| Sensitivity % | 45.8 | 15.0 | 0.001 | | 52.8 | 30.7 | <0.001 | |
| (95%CI), n/N | (37.2–54.7), 55/120 | (9.7–22.5), 18/120 | | | (44.1–61.2), 67/127 | (23.4–39.2), 39/127 | | |
| Specificity% | 95.2 | 100.0 | 0.029 | | 100.0 | 99.1 | 1.000 | |
| (95% CI), *n/N* | (89.8–97.8), 118/124 | (97.0–100.0), 124/124 | | 0.456 | (96.5–100.0), 107/107 | (94.9–99.8), 106/107 | | 0.498 |
| PPV% | 90.2 | 100.0 | 0.328 | | 100.0 | 97.5 | 0.373 | |
| (95% CI), *n/N* | (80.2–95.4), 55/61 | (82.4–100.0), 18/18 | | | (94.6–100.0), 67/67 | (87.1–99.6), 39/40 | | |
| NPV% | 64.5 | 54.9 | 0.055 | | 64.1 | 54.6 | 0.086 | |
| (95% CI), *n/N* | (57.3–71.1), 118/183 | (48.3–61.2), 124/226 | | | (56.6–71.0), 107/167 | (47.6–61.5), 106/194 | | |

[a]P values for sensitivity, specificity, PPV, and NPV were compared between AIMLAM and Xpert using McNemar's paired test.
[b]P values for sensitivity, specificity, PPV, and NPV were compared between AIMLAM and Ultra using McNemar's paired test.

larger sample sizes are required to evaluate AIMLAM testing in children across diverse settings.

## Conclusion

The overall sensitivity of AIMLAM was moderate, whereas a higher sensitivity was observed in patients with bacteriologically confirmed and severe TB. Age-specific cutoff values may be needed to optimize the diagnostic model in children. Our findings provide preliminary evidence for a method of diagnosing ATB in pediatric patients with samples that are difficult to obtain.

## ACKNOWLEDGMENTS

We gratefully acknowledge all children who participated in the study and provided samples.

This study was sponsored by grants from the National Natural Science Foundation of China (82470011, 82170007), the Beijing Natural Science Foundation (L246011), Training Plan for High-Level Public Health Technical Talents of Beijing Municipal Health Commission (2022-02-04), and Funding for Reform and Development of Beijing Municipal Health Commission (EYGF-HX-05).

L.S. and H.Z. designed the study. H.L., M.F., and L.D. enrolled the subjects. Y.C., J.K., and J.L. performed the test and collected the clinical data. Y.C., J.X., and L.S. analyzed the data.

Y.C., L.S., and H.Z. wrote and revised the manuscript. All authors reviewed the results and contributed to the manuscript. All authors approved the final version of the manuscript.

The funder of this study had no role in study design, data collection, data analysis, data interpretation, writing of the report, or in the decision to submit for publication. The authors had full access to the data and made the decision to publish the manuscript.

## AUTHOR AFFILIATIONS

[1]Laboratory of Respiratory Diseases, Beijing Key Laboratory of Core Technologies for the Prevention and Treatment of Emerging Infectious Diseases in Children, Key Laboratory of Major Diseases in Children, Ministry of Education, Beijing Pediatric Research Institute, Beijing Children's Hospital, Capital Medical University, National Clinical Research Center for Respiratory Diseases, National Center for Children's Health, Beijing, China

[2]Department of Pediatric Pulmonology, The Second Affiliated Hospital and Yuying Children's Hospital of Wenzhou Medical University, Wenzhou, China

[3]Department of Science and Education, Xinjiang Institute of Pediatrics, Xinjiang Hospital of Beijing Children's Hospital, Children's Hospital of Xinjiang Uygur Autonomous Region, The Seventh People's Hospital of Xinjiang Uygur Autonomous Region, Urumqi, China

[4]The Seventh People's Hospital of Liangshan Yizu Autonomous Prefecture, Liangshan, Sichuan, China

[5]The No. 1 People's Hospital of Liangshan Yuzu Autonomous Prefecture, Liangshan, Sichuan, China

## AUTHOR ORCIDs

Hailin Zhang ⓘ http://orcid.org/0000-0002-7663-0418
Lin Sun ⓘ http://orcid.org/0000-0003-1044-2470

## AUTHOR CONTRIBUTIONS

Yiyi Chen, Data curation, Writing - original draft | Haiyan Li, Investigation | Jing Xiao, Methodology | Hui Qi, Project administration | Jie Kang, Formal analysis | Qifeng Li, Validation | Jingjing Li, Resources | Min Fang, Validation | Li Duan, Visualization | Xiaomeng Wu, Visualization | Hailin Zhang, Writing – review and editing | Lin Sun, Writing – review and editing

## ETHICS APPROVAL

This study was approved by the Ethics Committee of Beijing Children's Hospital (2025-Y-040-D), and the guardians of all participants signed written informed consent.

## ADDITIONAL FILES

The following material is available online.

### Supplemental Material

**Supplemental figure and table (Spectrum03571-25-s0001.pdf).** Figure S1 and Tables S1 to S5.
**Supplemental material (Spectrum03571-25-s0002.docx).** STARD check list.

### Open Peer Review

**PEER REVIEW HISTORY (review-history.pdf).** An accounting of the reviewer comments and feedback.

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
