## [Reviewer comments · Microbiology Spectrum]

Microbiology Spectrum

Efficacy of a Chemiluminescence-Based Urinary LAM(AIMLAM) Assay in diagnosis of active TB in Chinese Children

Yiyi Chen, Haiyan Li, Jing Xiao, Hui Qi, Jie Kang, Qifeng Li, Jingjing Li, Min Fang, Li Duan, Xiaomeng Wu, Hailin Zhang, and Lin Sun

Corresponding Author(s): Lin Sun, Beijing Children's Hospital Capital Medical University

Review Timeline:

Submission Date:	November 12, 2025
Editorial Decision:	December 29, 2025
Revision Received:	January 10, 2026
Accepted:	February 10, 2026

Editor: Yuan Pin Hung

Reviewer(s): Disclosure of reviewer identity is with reference to reviewer comments included in decision letter(s). The following individuals involved in review of your submission have agreed to reveal their identity: Tabinda Anwar (Reviewer #1); Andriansjah Andriansjah (Reviewer #3)

Transaction Report:

DOI: <https://doi.org/10.1128/spectrum.03571-25>

Re: Spectrum03571-25 (**Efficacy of a Chemiluminescence-Based Urinary LAM(AIMLAM) Assay in diagnosis of active TB in Chinese Children**)

Dear Dr. Lin Sun:

Thank you for the privilege of reviewing your work. Below you will find my comments, instructions from the Spectrum editorial office, and the reviewer comments.

Revision Guidelines

Sincerely,
Yuan Pin Hung
Editor
Microbiology Spectrum

Reviewer #1 (Comments for the Author):

This manuscript addresses an important gap in pediatric tuberculosis diagnosis by evaluating the AIMLAM urine test in a large cohort of children. The study is well-structured, with rigorous patient classification, clear methodology, and thorough statistical analysis. The results demonstrate moderate overall sensitivity of AIMLAM, with higher sensitivity in bacteriologically confirmed, severe, and extrapulmonary TB cases, highlighting its utility in difficult-to-diagnose pediatric patients. The combined use of

AIMLAM with molecular tests significantly improved diagnostic sensitivity, which has practical implications, especially in resource-limited settings. Limitations include the lack of direct comparison with other LAM kits, small sample sizes for certain TB subtypes, and absence of NTM evaluation. Overall, the study provides meaningful evidence supporting AIMLAM as a non-invasive adjunctive diagnostic tool in childhood TB.

Suggestions:

The manuscript does not address multidrug-resistant tuberculosis (MDR-TB). Although Xpert/Xpert Ultra assays were performed, rifampicin resistance results were not reported, and patients were not stratified according to drug-resistance status. Given the growing burden of pediatric MDR-TB and the potential advantage of LAM-based diagnostics being independent of drug susceptibility, this represents an important gap. The authors should explicitly acknowledge this limitation and discuss the potential role of AIMLAM in MDR-TB case detection.

Reviewer #3 (Comments for the Author):

The research conducted by the author is very important for solving the problem of limited adequate samples for TB diagnosis, especially in pediatric patients. Information related to alternative TB diagnostic tools will assist in case detection and accurate patient treatment efforts.

However, the reviewer believes it is important to verify a few of the following points.

1. In lines 236-238, it is stated that a Sankey diagram (Figure 2) shows that out of 193 (33.3%, 193/579) positive AIMLAM results, 29 were attributed to BC-TB, 144 to CD-TB, and 20 to the DC (80.0%, 16/20) and HC (20.0%, 4/20) groups, but this data is not detailed in Figure 2. Can this diagram be improved?
2. In Figure 1, why does the scheme show that the determination of BC and CD-TB uses TB LAM instead of culture as the gold standard? The materials and methods section already describes the culture procedure. Can you clarify where the culture process is indicated in Figure 1? Also, is the TB LAM mentioned here the same as AIMLAM?
3. In Table 2, what is AIMLAM's accuracy compared to? The reviewer recommends that the authors enhance the table title to include more specific details.
4. In lines 304-306, why does the Ultra have lower sensitivity compared to Xpert? As we know, Ultra's LOD is lower than Xpert's. Did the author work with the same sample in the same buffer for both methods? A different result was found in Table 3, where Ultra's sensitivity was higher than Xpert's. Could the author explain this?

I also give some notes on the PDF file.

**Title Page**

**Efficacy of a Chemiluminescence-Based Urinary LAM(AIMLAM) Assay in diagnosis of**
**active TB in Chinese Children**

Yiyi Chen¹, Haiyan Li², Jing Xiao¹, Hui Qi¹, Jie Kang¹, Qifeng Li⁵, Jingjing Li¹, Min Fang^{3,4},
Li Duan^{3,4}, Xiaomeng Wu¹, Hailin Zhang^{2*}, Lin Sun^{1*}

1 Laboratory of Respiratory Diseases, Beijing Pediatric Research Institute, Beijing Children's
Hospital, Capital Medical University, Beijing Key Laboratory of Core Technologies for the
Prevention and Treatment of Emerging Infectious Diseases in Children, Key Laboratory of
Major Diseases in Children, Ministry of Education, National Clinical Research Center for
Respiratory Diseases, National Center for Children's Health, Beijing, China

2 Department of Pediatric Pulmonology, The Second Affiliated Hospital and Yuying
Children's Hospital of Wenzhou Medical University, Wenzhou 325027, China

3 The Seventh People's Hospital of Liangshan Yizu Autonomous Prefecture, Liangshan,
Sichuan, China

4 The No. 1 People's Hospital of Liangshan Yuzu Autonomous Prefecture, Liangshan,
Sichuan, China

5 Department of Science and Education, Xinjiang Institute of Pediatrics, Xinjiang Hospital of
Beijing Children's Hospital, Children's Hospital of Xinjiang Uygur Autonomous Region, The
Seventh People's Hospital of Xinjiang Uygur Autonomous Region, Urumqi 830054, China.

*** Correspondence Authors:**

Prof Lin Sun, Laboratory of Respiratory Diseases, Beijing Pediatric Research Institute,
Beijing Children's Hospital, Capital Medical University, Beijing Key Laboratory of Core
Technologies for the Prevention and Treatment of Emerging Infectious Diseases in Children,
Key Laboratory of Major Diseases in Children, Ministry of Education, National Clinical
Research Center for Respiratory Diseases, National Center for Children's Health, Beijing,
China. No. 56 Nan-li-shi Road, 100045 Beijing, China. Email: sunlinbch@163.com.

Prof Hailin Zhang, Department of Pediatric Respiratory Medicine, The Second Affiliated
Hospital and Yuying Children's Hospital of Wenzhou Medical University, Wenzhou 325027,
China. No.109, Academy West Road, Lucheng District, Wenzhou, China. Email:
zhlwz97@hotmail.com.

**Chemiluminescence-based urinary lipoarabinomannan (AIMLAM) assay for**
**diagnosing active TB in Chinese children**

Yiyi Chen¹, Haiyan Li², Jing Xiao¹, Hui Qi¹, Jie Kang¹, Jingjing Li¹, Min Fang^{3,4}, Li Duan^{3,4},
Xiaomeng Wu¹, Hailin Zhang^{2*}, Lin Sun^{1*}

**Abstract**

Background. Child-friendly triage tests that do not rely on sputum for diagnosing active
tuberculosis (ATB) in children are urgently required.

Method. This study aimed to assess a chemiluminescence-based urinary lipoarabinomannan
test (AIMLAM) for diagnosing tuberculosis (TB) in children in China.

Results. Among the 579 children enrolled, 331 had TB, 169 had infectious diseases, and 79
were healthy controls. The AUC of the AIMLAM test for distinguishing TB from non-TB
(children with infectious diseases and healthy controls) was 0.74 (95% CI, 0.70–0.78), with a
sensitivity of 52.2% (95% CI, 46.6%–57.6%) and a specificity of 91.9% (95% CI, 87.6%–
94.9%). In 288 children with clinically diagnosed TB, the sensitivity of the test was 50.0%
(95% CI, 44.1%–55.9%). LAM concentrations in extrapulmonary TB were significantly
higher than those in pulmonary TB ($P=0.003$), which resulted in a higher sensitivity of 63.6%
(95% CI, 54.7%–72.1%). The diagnostic sensitivity of AIMLAM was significantly superior
to that of Xpert ($P=0.004$) and Ultra ($P=0.002$). Combining diagnostics expanded the
decision curve analysis thresholds while improving the cost-benefit ratio.

Conclusions. The sensitivity of AIMLAM is moderate but highest in extrapulmonary TB
patients. The test appears to be promising for the rapid diagnosis of TB in children, especially

in those with negative bacteriological results.

Keywords. Pediatric tuberculosis; diagnostics; AIMLAM; joint diagnosis; clinical practice

**Importance**

This study of a large-sample cohort of children in China assessed the diagnostic accuracy of
the AIMLAM test kit as an auxiliary tool for childhood TB, and to determine an age-specific
reference interval for LAM concentrations in children. Our findings showed that the overall
sensitivity of AIMLAM was moderate, while a higher sensitivity was observed in patients
with bacterially confirmed and severe TB. Age-specific cutoff values may be needed to
optimize the diagnostic model in children. These results provide preliminary evidence for a
method of diagnosing ATB (an infectious state of MTB actively replicating in the host,
causing clinical symptoms, tissue and organ damage, and inflammatory responses.)
[revised manuscript text omitted]

Group	Sensitivity%	P value	PPV%	P value	NPV%	P value	Specificity%	253	CI, confide nce interval; PPV, positive predicti ve value; NPV, negativ e predicti ve value.
	(95%CI), n/N		(95%CI), n/N		(95%CI), n/N		(95%CI), n/N	P value	
Active TB	52.2		89.6		59.1		91.9	256	
(N=331)	(46.6-57.6),173/331		(83.6-93.0),173/193		(54.0-64.0),228/386		(87.6-94.9),228/248	257	
≤ 5 years	62.9		80.0		75.7		88.0	258	
	(50.4-73.9),44/70		(66.7-89.1),44/55		(66.2-83.2),81/107		(79.2-93.6),81/92	259	
5~10 years	46.8	<0.001	92.9	0.938	61.4	0.868	95.9	260	
	(37.4-56.5),52/111		(81.9-97.7),52/56		(53.2-69.1),94/153		(89.3-98.7),94/98	261	
>10 years	51.3		93.9		42.1		91.4	262	
	(43.1-59.5),77/150		(85.7-97.7),77/82		(33.4-51.2),53/126		(80.3-96.8),53/58	263	
Bacteriologically confirmed TB									
(BC-TB)	67.4		59.2		94.2			264	
(N=43)	(51.3-80.5),29/43	0.049	(43.3-71.5),29/49	<0.001	(90.3-96.7),228/242	0.557		265	
Clinical diagnosed TB (CD-TB)	50.0		87.8		61.3			266	
(N=288)	(44.1-55.9),144/288		(89.0-91.8),144/164		(56.1-66.2),228/372				
Severe TB	69.4		62.9		93.8		91.9	1.000	
(N=49)	(54.4-81.3),34/49	0.011	(48.7-75.4),34/54	<0.001	(89.8-96.4),228/243	<0.001	(87.6-94.9),228/248		
Mild TB	49.3		87.4		61.5				
(N=282)	(43.3-55.5),139/282		(81.0-92.0),139/159		(56.3-66.4),228/371				
Pulmonary TB	46.0		83.1		66.5				
(N=213)	(39.2-52.9),98/213	0.003	(74.8-89.1),98/118	0.558	(61.2-71.4),228/343	<0.001			
Extrapulmonary TB	63.6		78.9		84.1				
(N=118)	(54.1-72.1),75/118		(69.1-86.3), 75/95		(79.1-88.2),228/271				

3.2.2 Best cutoff value of AIMLAM in the diagnosis of TB in children

The optimal thresholds of AIMLAM was then analyzed to achieve the best diagnostic
accuracy in children. The optimal positive threshold for discriminating total ATB from
non-TB controls were 0.1, 0.13 and 0.14, respectively. (Figure 3A-G). Firstly, the DC and HC
groups were merged as the non-TB controls. A moderate accuracy of AIMLAM was observed
in diagnosing the total ATB, with an AUC of 0.74 (95% CI, 0.70–0.78). A similar diagnostic
accuracy was observed in BC-TB (AUC, 0.82; 95% CI, 0.72–0.91) and CD-TB (AUC,0.74;
95% CI, 0.69–0.79), respectively (Figure 3A–C). Either comparing with healthy children
(Figure 3D–F) or those with infectious diseases (Figure 3G–I), the diagnostic accuracy of
AIMLAM was similar in diagnosing ATB.

The diagnostic accuracy metrics of AIMLAM distinguishing ATB from Non-TB were further
validated at thresholds of 0.14, 0.13, and 0.1 using Bootstrapping. The corrected sensitivities
were 67.4% (95% CI, 62.2%-72.4%), 68.5% (95% CI, 63.3%-73.7%), and 70.8% (95% CI,
65.4%-76.0%), respectively. The specificity values were 80.6% (95% CI, 75.5%-85.2%),
79.5% (95% CI, 74.4%-84.7%), and 77.5% (95% CI, 71.9%-82.5%), respectively.(Table S1,
S5).

3.3 Effectiveness of AIMLAM across various TB subgroups

Among the 331 children with ATB, 49 had severe TB and 282 had mild TB. A violin plot
showed that data density in the mild TB group was mainly observed in the low value regions,
and showed a tighter distribution than that in the severe group. The median concentration of
AIMLAM in the mild group was 0.28 U/ml (IQR: 0.10–0.86), which was lower than that

(0.68 U/ml, IQR: 0.16–2.21) in the severe group. Nevertheless, this difference was not
significant, which may have been due to more dispersed data in the severe group (Figure 4A).
In addition, 213 children had pulmonary TB, and 118 were diagnosed with extrapulmonary
TB. The median AIMLAM concentration in children with extrapulmonary TB (0.85, IQR:
0.16–2.17 U/ml) was significantly higher than that in those with pulmonary TB (0.47, IQR:
0.1–2.17 U/ml) ($P=0.013$, Figure 4B).

Among the 118 children with extrapulmonary TB, the main types were abdominal TB and
tuberculous peritonitis (44.1%) and tuberculous meningitis (33.9%) (Figure 4C). Abdominal
TB and tuberculous peritonitis frequently presented in conjunction with other forms of
extrapulmonary TB, which contributed to the elevated detection rate with AIMLAM (37/52,
71.2%) (Figure 4D). The sensitivity of AIMLAM for tuberculous meningitis and bone TB
was similar (65%, 26/40 vs. 73.3%, 11/15). All five cases with urinary tract TB had positive
results (100.0%). By contrast, tuberculous pericarditis (40.0% (4/10) and lymph node TB
(46.7%, 7/15) showed the lowest detection rates.

To compare the differences between AIMLAM and molecular tests in diagnosing
extrapulmonary and pulmonary TB, we also examined the diagnostic performance of Xpert
and Ultra in intrapulmonary and extrapulmonary cases (Table S2). The data indicated that,
relative to extrapulmonary TB (38.3%; 95% CI, 24.9%-53.6%), Xpert demonstrated a
significantly higher sensitivity in pulmonary TB (86.9%; 95% CI, 78.2%-92.5%). Conversely,
Ultra's sensitivity in extrapulmonary TB was 47% (95% CI, 34.7%-69.6%), which was nearly

[revised manuscript text omitted]

**Author contributions**

LS and HZ designed the study. HL, MF and LD enrolled the subjects. YC, JK and JL
performed the test and collected the clinical data. YC, JX and LS analyzed the data. YC, LS
and HZ wrote and revised the manuscript. All authors reviewed the results and contributed to
the manuscript. All authors approved the final version of the manuscript.

**Disclaimer**

The funder of this study had no role in study design, data collection, data analysis, data

interpretation, writing of the report, or in the decision to submit for publication. The authors
had full access to the data and made the decision to publish the manuscript.

**Financial support**

This study was sponsored by grants from National Natural Science Foundation of China
(82470011, 82170007), the Beijing Natural Science Foundation (L246011), Training Plan for
High-Level Public Health Technical Talents of Beijing Municipal Health Commission
(2022-02-04) and Funding for Reform and Development of Beijing Municipal Health
Commission (EYGF-HX-05).

**Acknowledgements**

We gratefully acknowledge all children who participated in the study and provided samples.

**Potential conflicts of interest**

The authors declare no competing interests.

**References**

- 1. Basnyat, B., M. Caws, and Z. Udawadia, *Tuberculosis in South Asia: a tide in the affairs of men.*
*Multidiscip Respir Med*, 2018. **13**: p. 10.
- 2. DiNardo, A.R., et al., *Culture is an imperfect and heterogeneous reference standard in pediatric*
*tuberculosis.* *Tuberculosis (Edinburgh, Scotland)*, 2016. **101S**: p. S105-S108.
- 3. Chu, P., et al., *Bacteremia tuberculosis among HIV-negative children in China.* *Pediatric Investigation*,
2022. **6**(3): p. 197-206.
- 4. Nicol, M.P., et al., *Accuracy of the Xpert MTB/RIF test for the diagnosis of pulmonary tuberculosis in*
*children admitted to hospital in Cape Town, South Africa: a descriptive study.* *The Lancet. Infectious*
*Diseases*, 2011. **11**(11): p. 819-824.
- 5. Nicol, M.P., et al., *Accuracy of Xpert Mtb/Rif Ultra for the Diagnosis of Pulmonary Tuberculosis in*
*Children.* *The Pediatric Infectious Disease Journal*, 2018. **37**(10): p. e261-e263.
- 6. Theron, G., et al., *False-Positive Xpert MTB/RIF Results in Retested Patients with Previous Tuberculosis:*
*Frequency, Profile, and Prospective Clinical Outcomes.* *Journal of Clinical Microbiology*, 2018. **56**(3).

- 7. Purohit, M. and T. Mustafa, *Laboratory Diagnosis of Extra-pulmonary Tuberculosis (EPTB) in*
*Resource-constrained Setting: State of the Art, Challenges and the Need*. Journal of Clinical and
Diagnostic Research : JCDR, 2015. **9**(4): p. EE01-EE06.
- 8. Gao, M., et al., *Advancements in LAM-based diagnostic kit for tuberculosis detection: enhancing TB*
*diagnosis in HIV-negative individuals*. Frontiers In Microbiology, 2024. **15**: p. 1367092.
- 9. Chan, J., et al., *Lipoarabinomannan, a possible virulence factor involved in persistence of*
*Mycobacterium tuberculosis within macrophages*. Infection and Immunity, 1991. **59**(5): p. 1755-1761.
- 10. Correia-Neves, M., et al., *Biomarkers for tuberculosis: the case for lipoarabinomannan*. ERJ Open
Research, 2019. **5**(1).
- 11. Bah, A., et al., *The Lipid Virulence Factors of Mycobacterium tuberculosis Exert Multilayered Control*
*over Autophagy-Related Pathways in Infected Human Macrophages*. Cells, 2020. **9**(3).
- 12. Hamasur, B., et al., *Rapid diagnosis of tuberculosis by detection of mycobacterial lipoarabinomannan in*
*urine*. Journal of Microbiological Methods, 2001. **45**(1): p. 41-52.
- 13. Lu, J., et al., *Clinical application of the urinary lipoarabinomannan (AIMLAM) test in PLHIV with TB*.
AIDS Research and Therapy, 2025. **22**(1): p. 59.
- 14. Sossen, B., et al., *Urine-Xpert Ultra for the diagnosis of tuberculosis in people living with HIV: a*
*prospective, multicentre, diagnostic accuracy study*. The Lancet. Global Health, 2024. **12**(12): p.
e2024-e2034.
- 15. Adams, J.D., et al., *Short-Term Stability of Urine Electrolytes: Effect of Time and Storage Conditions*.
International Journal of Sport Nutrition and Exercise Metabolism, 2022. **32**(2): p. 111-113.
- 16. Esmail, A., et al., *An Optimal Diagnostic Strategy for Tuberculosis in Hospitalized HIV-Infected Patients*
*Using GeneXpert MTB/RIF and Alere Determine TB LAM Ag*. Journal of Clinical Microbiology, 2020.
**58**(10).
- 17. Bjerrum, S., et al., *Lateral flow urine lipoarabinomannan assay for detecting active tuberculosis in*
*people living with HIV*. The Cochrane Database of Systematic Reviews, 2019. **10**(10): p. CD011420.
- 18. Broger, T., et al., *Novel lipoarabinomannan point-of-care tuberculosis test for people with HIV: a*
*diagnostic accuracy study*. The Lancet. Infectious Diseases, 2019. **19**(8): p. 852-861.
- 19. Huang, L., et al., *Diagnostic value of chemiluminescence for urinary lipoarabinomannan antigen assay*
*in active tuberculosis: insights from a retrospective study*. Frontiers In Cellular and Infection
Microbiology, 2023. **13**: p. 1291974.
- 20. Peng, L., et al., *Developing a method to detect lipoarabinomannan in pleural fluid and assessing its*
*diagnostic efficacy for tuberculous pleural effusion*. Heliyon, 2023. **9**(8): p. e18949.
- 21. Organization, W.H., *Global tuberculosis report 2024*. 2024.
- 22. *WHO consolidated guidelines on tuberculosis: Module 3: diagnosis – rapid diagnostics for tuberculosis*
*detection*. 2021, Geneva: World Health Organization.
- 23. Graham, S.M., et al., *Clinical Case Definitions for Classification of Intrathoracic Tuberculosis in Children:*
*An Update*. Clinical Infectious Diseases : an Official Publication of the Infectious Diseases Society of
America, 2015. **61**Suppl 3(Suppl 3): p. S179-S187.
- 24. Opota, O., et al., *Added value of molecular assay Xpert MTB/RIF compared to sputum smear*
*microscopy to assess the risk of tuberculosis transmission in a low-prevalence country*. Clinical
Microbiology and Infection : the Official Publication of the European Society of Clinical Microbiology
and Infectious Diseases, 2016. **22**(7): p. 613-619.
- 25. Detjen, A.K., et al., *Xpert MTB/RIF assay for the diagnosis of pulmonary tuberculosis in children: a*
*systematic review and meta-analysis*. The Lancet. Respiratory Medicine, 2015. **3**(6): p. 451-461.

**Figure 1.** Flow chart of the study population. TB, tuberculosis; DCs, Infectious disease
Control; HC, Healthy Control; BC-TB, Bacteriologically confirmed TB; CD-TB, Clinically
diagnosed TB; AIMLAM, Chemiluminescence-Based Urinary LAM; Xpert, Xpert MTB/RIF;
and Ultra, Xpert MTB/RIF Ultra.

**Figure 2.** Sankey diagram of the distribution of population flows for AIMLAM diagnosis.

**Figure 3.** ROC curve of AIMLAM for the detection of TB. (A-C) ROC curves of AIMLAM
in different age groups to distinguish between non-TB and TB populations(A: Active TB, B:
BC-TB, C: CD-TB), including the area under the ROC curves(AUCs) and the overall optimal
cut-off value; (D-F) ROC curves of AIMLAM in different age groups to distinguish between
TB populations(A: Active TB, B: BC-TB, C: CD-TB) and HC populations, including the area
under the ROC curves(AUCs) and the overall optimal cut-off value; (G-I) ROC curves of
AIMLAM in different age groups to distinguish between TB populations(A: Active TB, B:
BC-TB, C: CD-TB) and DC populations, including the area under the ROC curves(AUCs)
and the overall optimal cut-off value.

**Figure4.** Comparison of AIMLAM test results in different subgroups. (A) Violin plot of the
difference in the diagnostic presentation of AIMLAM in mild TB and severe TB. (B) Violin
plot of the difference in the diagnostic presentation of AIMLAM in Pulmonary TB and
Extrapulmonary TB (C) Distribution of LAM-positive cases in different types of
extrapulmonary tuberculosis. (D) Diagnostic efficacy of AIMLAM in different
extrapulmonary tuberculosis. Colored squares represented LAM-positive results and gray
squares represented LAM-negative results.

**Figure 5.** Performance of different assays and Decision Curve Analysis (DCA) of AIMLAM
and other assays. (A) Venn diagram of AIMLAM and Xpert in participants classified as active
TB (N = 270). (B) Results profile of the detection of 252 participants with different TB
classifications via AIMLAM and Xpert. (C) Results profile of the detection of 270
participants with different TB classifications via AIMLAM, Xpert, IGRA, and TST.
(D)Venn diagram of AIMLAM and Ultra in participants classified as active TB (N = 252). (E)
Table of AIMLAM and Xpert in participants classified as active TB, BC-TB and CD-TB
(N = 270). (F) Results profile of the detection of 252 participants with different TB
classifications via AIMLAM, Ultra, IGRA, and TST.

Responses to Reviewers' Comments

Dear editor and reviewers,

We thank you very much for the constructive comments on our manuscript entitled “**Efficacy of a Chemiluminescence-Based Urinary LAM(AIMLAM) Assay in diagnosis of active TB in Chinese Children**”. We have revised the manuscript, and we greatly appreciate the improvement in our manuscript due to the comments of the reviewers. We highlighted all the revisions in yellow. Our point-to-point responses to the queries raised by the reviewers are listed as follows:

Responses to the reviewers' comments: all numbers refer to the revised version

Reviewer #1

1. The manuscript does not address multidrug-resistant tuberculosis (MDR-TB). Although Xpert/Xpert Ultra assays were performed, rifampicin resistance results were not reported, and patients were not stratified according to drug-resistance status. Given the growing burden of pediatric MDR-TB and the potential advantage of LAM-based diagnostics being independent of drug susceptibility, this represents an important gap. The authors should explicitly acknowledge this limitation and discuss the potential role of AIMLAM in MDR-TB case detection.

Response: Thank you for this insightful and critical comment. We fully agreed with your opinion. In the present study, a total of 6 children were confirmed to be rifampicin-resistant by Xpert and Ultra. The sample size for rifampicin-resistance statuses was limited for subsequent analysis, which was added as a limitation in the discussion (Lines 392-401).

“What’s more, given the growing burden of pediatric multidrug-resistant tuberculosis (MDR-TB), we attempted to analyze the accuracy of AIMLAM detection in MDR-TB populations. In the present study, a total of 6 children were confirmed to be rifampicin-resistant by Xpert and Ultra. The sample size for rifampicin-resistance statuses was limited for subsequent analysis. A recent study revealed that Mannose-capped lipoarabinomannan (ManLAM)-related genes (pimB, mptA, dprE1, and embC) were most

up-regulated under isoniazid (INH) treatment, indicating their involvement in drug resistance and MTB adaptability through ManLAM modulation[1]. This suggests that the detection of AIMLAM or its related genes might play a role in the diagnosis of MDR-TB, which needs to be furtherly validated with a large sample size.”

Reviewer #2

1. In lines 236-238, it is stated that a Sankey diagram (Figure 2) shows that out of 193 (33.3%, 193/579) positive AIMLAM results, 29 were attributed to BC-TB, 144 to CD-TB, and 20 to the DC (80.0%, 16/20) and HC (20.0%, 4/20) groups, but this data is not detailed in Figure 2. Can this diagram be improved?

Response: We appreciate the reviewer’s valuable suggestion. We have revised the Sankey diagram (Figure 2).

2. In Figure 1, why does the scheme show that the determination of BC and CD-TB uses TB LAM instead of culture as the gold standard? The materials and methods section already describes the culture procedure. Can you clarify where the culture process is indicated in Figure 1? Also, is the TB LAM mentioned here the same as AIMLAM?

Response : We appreciate the reviewer for raising this important point. The revisions are as follows: (i) as stated in Methods (Lines 145-147), we differentiated between BC-TB and CD-TB based on positive MTB culture or acid-fast staining as the gold standard. Figure 1 showed that we tested the entire included population with AIMLAM without initial discrimination, then categorized the cohort into Active TB (ATB), disease control (DC), and healthy control (HC) group according to diagnostic criteria(Lines 145-165), rather than dividing the cohort solely based on AIMLAM results. (ii) The culture process was indicated when classifying ATB into BC-TB and CD-TB. (iii) TB LAM refers to AIMLAM mentioned in the text, and this has been revised in Figure 1.

3. In Table 2, what is AIMLAM's accuracy compared to? The reviewer recommends that the authors enhance the table title to include more specific details.

Response: Thank you for your valuable suggestion. We stratified Active TB children into subgroups based on the age, diagnostic criteria, disease severity, and location of occurrence. The accuracies of AIMLAM were compared within (i) three age subgroups, (ii) BC-TB and CD-TB subgroups, (iii) severe and mild TB subgroups, (iv) intrapulmonary and extrapulmonary TB subgroups. In addition, we revised the title of Table 2, and now it reads: “The accuracy of AIMLAM in children categorized by age, diagnostic criteria, disease severity, and location of occurrence.” (Lines 248)

4. In lines 304-306, why does the Ultra have lower sensitivity compared to Xpert? As we know, Ultra's LOD is lower than Xpert's. Did the author work with the same sample in the same buffer for both methods? A different result was found in Table 3, where Ultra's sensitivity was higher than Xpert's. Could the author explain this?

Response : Thank you for your valuable comments. In response, we rechecked the data and discovered a miscalculation in Table S2 regarding the Xpert data. We corrected and updated the table, showing that Ultra's sensitivity is higher than Xpert, which aligns with the results in Table 3. We also made revisions to the original text, and now the text states: “The data showed that, compared to pulmonary TB (27.8%; 95% CI, 18.6%-39.2%), Ultra demonstrated a much higher sensitivity for extrapulmonary TB at 47% (95% CI, 34.7%-69.6%). Likewise, Xpert's sensitivity for extrapulmonary TB was 38.3% (95% CI, 24.9%-53.6%), which was more than twice as high as its sensitivity for pulmonary TB at 13.1% (95% CI, 7.5%-21.8%).”(Lines 286-290)

5. “We selected the midstream urine and avoided using samples with high proteinuria and fatty urine (proteinuria were urine protein levels exceeding 100 mg/L or 150 mg/24 hours; fatty urine were cloudy urine with floating oil droplets, increased foam, free fat, oval fat bodies, and small fat tubular bodies).” Notes: In routine applications, would these factors serve as exclusion criteria for a sample?

Response : Thank you for pointing this out. Proteinuria and fatty urine should be excluded when performing AIMLAM testing, and we have added this rule in the exclusion criteria. The

text now reads: “Children with incomplete information, HIV-positive results, and samples with high proteinuria and fatty urine were excluded from this study.” (Lines 163-165).

6. “Significant variations in sensitivity were observed across different age groups, with a sensitivity of 62.9% (95% CI, 50.4–73.9%) in children younger than 5 years of age, 46.8% (95% CI, 37.4–56.5%) in those 5–10 years of age, and 51.3% (95% CI, 43.1–59.5%) in those older than 10 years of age.” Notes: It is better to include this in the same paragraph as the previous sentence since it still pertains to Table 2.

Response : Thank you for your valuable suggestions. We have combined this paragraph with the previous one into one paragraph. (Lines 233-247)

7. In Table 3, Xpert and Ultra compared to what?

Response: We appreciate the reviewer’s careful attention to detail. We have revised the table and added the concrete comparison in the legend (Table 3).

1. Yimcharoen, M., et al., *The Regulation of ManLAM-Related Gene Expression in Mycobacterium tuberculosis with Different Drug Resistance Profiles Following Isoniazid Treatment*. Infection and Drug Resistance, 2022. **15**: p. 399-412.

Re: Spectrum03571-25R1 (**Efficacy of a Chemiluminescence-Based Urinary LAM(AIMLAM) Assay in diagnosis of active TB in Chinese Children**)

Dear Dr. Lin Sun:

Your manuscript has been accepted, and I am forwarding it to the ASM production staff for publication. Your paper will first be checked to make sure all elements meet the technical requirements. ASM staff will contact you if anything needs to be revised before copyediting and production can begin. Otherwise, you will be notified when your proofs are ready to be viewed.

Sincerely,
Yuan Pin Hung
Editor
Microbiology Spectrum